# Innovative machine learning approach and evaluation campaign for predicting the subjective feeling of work-life balance among employees

**Aleksandra Pawlicka** [1]*, **Marek Pawlicki**[2], **Renata Tomaszewska**[3], **Michał Choraś**[4], **Ryszard Gerlach**[5]

**1** ITTI, Bydgoszcz, Poland, **2** UTP University of Science and Technology, Bydgoszcz, Poland, **3** Kazimierz Wielki University in Bydgoszcz, Bydgoszcz, Poland, **4** UTP University of Science and Technology, Bydgoszcz, Poland, **5** Kazimierz Wielki University in Bydgoszcz, Bydgoszcz, Poland

* oliko.aleksandra@gmail.com

## Abstract

At present, many researchers see hope that artificial intelligence, machine learning in particular, will improve several aspects of the everyday life for individuals, cities and whole nations alike. For example, it has been speculated that the so-called machine learning could soon relieve employees of part of the duties, which may improve processes or help to find the most effective ways of performing tasks. Consequently, in the long run, it would help to enhance employees' work-life balance. Thus, workers' overall quality of life would improve, too. However, what would happen if machine learning as such were employed to try and find the ways of achieving work-life balance? This is why the authors of the paper decided to utilize a machine learning tool to search for the factors that influence the subjective feeling of one's work-life balance. The possible results could help to predict and prevent the occurrence of work-life imbalance in the future. In order to do so, the data provided by an exceptionally sizeable group of 800 employees was utilised; it was one of the largest sample groups in similar studies in Poland so far. Additionally, this was one of the first studies where so many employees had been analysed using an artificial neural network. In order to enable replicability of the study, the specific setup of the study and the description of the dataset are provided. Having analysed the data and having conducted several experiments, the correlations between some factors and work-life balance have indeed been identified: it has been found that the most significant was the relation between the feeling of balance and the actual working hours; shifting it resulted in the tool predicting the switch from balance to imbalance, and vice versa. Other factors that proved significant for the predicted WLB are the amount of free time a week the employee has for themselves, working at weekends only, being self-employed and the subjective assessment of one's financial status. In the study the dataset gets balanced, the most important features are selected with the selectKbest algorithm, an artificial neural network of 2 hidden layers with 50 and 25 neurons, ReLU and ADAM is constructed and trained on 90% of the dataset. In tests, it predicts WLB based on the prepared dataset and selected features with 81% accuracy.

**Data Availability Statement:** The data is available in the Harvard Dataverse Repository: Renata Tomaszewska, 2020, "Work-life Balance - 800

workers", https://doi.org/10.7910/DVN/LTEIBX, Harvard Dataverse, V1, UNF:6: tJJB1wmDhLrDbPKnCNgOUQ== [fileUNF].

**Funding:** The author(s) received no specific funding for this work. Although one of the author's affiliation is commercial, this organization did not play any role in any part of the study; it only provided financial support in the form of author's salaries earned for completing the tasks completely unrelated to this study.

**Competing interests:** This affiliation does not alter our adherence to all PLOS ONE policies on sharing data and materials and we have no conflicts of interest to disclose.

## Introduction

At present, the notion of "artificial intelligence" (also called "machine-"or "deep learning") still triggers two kinds of reactions. On the one hand, some people immediately think of the science-fiction movies, where AI often dominates the world and strives for the annihilation of mankind. On the other hand, the ones who have slightly more understanding of the subject see hope that machine learning (ML) will improve several aspects of the everyday life for individuals, cities and whole nations alike. In fact, the advent of Industry 4.0 has already changed the way many professionals work. Although there are sceptics who prophesize that global unemployment caused by robotization is looming on the horizon, the shift towards a greater focus on workers' personal lives without compromising work commitments has already been observed [1]. For example, it has been speculated that the so-called machine learning could soon relieve employees of part of the duties, which may improve processes or help to find the most effective ways of performing tasks [2]. Consequently, in the long run, it would help to enhance employees' work-life balance. Thus, workers' overall quality of life would improve, too [3]. However, what would happen if machine learning as such were employed to try and find the ways of achieving work-life balance? The available subject literature does not provide many examples of similar trials. This is why the authors of the paper decided to utilize a machine learning tool to search for the factors that influence the subjective feeling of one's work-life balance. The possible results could help to predict and prevent the occurrence of work-life imbalance in the future. The correlations between some factors and work-life balance have indeed been found after having analysed the data provided by a group of 800 employees. The information came from a 2017 study conducted in Poland; the study was exceptionally large when compared to both the studies conducted in the country before and the work-life balance studies described in the literature in general.

The major contributions of this paper are: the novel methodology towards work-life balance studies with the support of machine learning tools, innovative experimental setup and interesting findings and results. The originality and novelty also lie in the fact that apart from evaluation studies (data from 800 subjects) the interdisciplinary methodology supported by neural networks is proposed and used in practice. In order to enable replicability of the study, the specific setup of the study and the description of the dataset are provided.

The paper is structured as follows: firstly, the idea of work-life balance along with its relation to the quality of life have been presented. Then, introductory information about neural networks has been provided. Then, the experimental setup and results of several innovative experiments have been given, followed by the final conclusions.

## Materials and methods

### Research questions

**Research question 1.** The main goal of this study was to find whether a machine learning tool is able to find any possible correlations between the employee-specific and workplace factors, and employees' subjective feeling of maintaining work-life balance.

**Research question 2.** Then, if any correlations were to be found, the following question was: Would shifting, increasing or decreasing any of the parameters result in achieving/ losing the balance? How?

### Background

In this section, the concept and definitions of work-life balance are presented. Then, the influence of the balance over the quality of life is discussed. Afterwards, the brief descriptions of

artificial neural networks are provided. Then, the attention is drawn to the common assumption regarding the relation between machine learning and work-life balance. We received information from 800 respondents, and several experiments have been conducted with the help of machine learning solutions to answer our research questions.

**The concept and definitions of work-life balance.** The debate over WLB has begun along with the major socio-economic changes: the increase in the number of female workers, Generations X and Y entering the labour market with new expectations, the technological advancements and the criticism of the so-called 'long-hour culture', i.e. forcing workers to work additional hours, regardless of the consequences for their family lives, health and overall wellbeing. It is not easy to give one, simple definition of work-life balance. Some researchers even call the nature of its meaning "problematic" [4]. The analysis of the subject literature allows finding several approaches to the idea of the balance; it is of complex and multi-faceted character [5]. Surprisingly, there is neither one clear definition of WLB, nor its measure [6].

Generally speaking, work-life balance (sometimes called work-family balance or simply WLB) is the state of comfortable equilibrium between an individual's priorities at work and in other aspects of their lives [7]. When the balance is maintained, the conflict between work and home is as slight as possible. This means that the work demands will not prevent the employee from gaining satisfaction from their personal life, whereas the aspects of their private lives do not spill over and exert an adverse impact on their work [8].

One of the most popular definition of WLB is the stance of David Clutterbuck, who claims that the balance between work and the life outside it is the state when an individual is able to manage the possible conflict between numerous demands on their time and energies in such a way, that their need for well-being and feeling fulfilment remains satisfied. In such a case, the concept of "balance" also encompasses stability and common sense, i.e. some subjective ideas of what is sensible, or what the personal equilibrium of the particular individual is. Thus, even when there occurs the conflict between work and personal activity, it does not necessarily mean the lack of balance. One may talk about the lack of balance if there arise the effects of the conflict, whether they be real or subjectively perceived ones. Clutterbuck points it out that achieving balance between work and private life comes down to adapting to the situation and dealing with it by first realizing the requirements concerning the investing of one's time and energies, then selecting one's values and making conscious choices based on them [9]. Sue Campbell Clark in turn sees "work-life balance" as good functioning and satisfaction at home and at work that a person achieves once they have minimized the conflict existing in both the spheres [10]. Greenhaus et al. have defined the conflict of multiple roles as: "Work-family balance reflects an individual's orientation across different life roles, an inter-role phenomenon." They have then defined work-life balance as "the extent to which an individual is engaged in– and equally satisfied with–his or her work role and family role." According to them, work-family balance consists of time balance, involvement balance, and satisfaction balance [11]. Other scientists, such as Kirchmeyer and Clark have primarily concentrated on the significance of individual satisfaction with multiple roles [10,12]. The concentrating upon individual satisfaction overlaps with the recognition that individuals perceive their multiple roles as varying in importance or salience to them; the salience of roles may change over time due to life changes (new baby, sickness, promotion, etc.) [6]. According to Greenhaus and Allen see WLB as "the extent to which an individual's effectiveness and satisfaction in work and family roles are compatible with the individual's life role priorities at a given point in time" [13]. Finally, Eby et al. claim that the research of work-life balance should focus on "whether one's expectations about work and family roles are met or not." [14] It is also wort mentioning that some researchers define work-life balance as the degree of autonomy one perceives oneself to have over the

demands of their multiple roles; the balance being "about people having a measure of control over when, where and how they work" [6,15].

All in all, although there were some attempts at measuring the objective work-life balance, most researchers agree this is no universal measure. On the contrary: it is something everyone may perceive in a different, personal way. Thus, it may be said that an individual maintains the balance between their work and professional life simply when they think and feel they do so.

All things considered, based on the most popular definitions of WLB, the authors have decided to perceive the personal, individual feeling of the worker as the most important factor determining their work-life balance level. Thus, this factor will be taken into account in the first place in the further part of this paper.

**The significant relation between work-life balance and the quality of life.** WLB is generally thought to promote well-being. In fact, it has been scientifically proved that there is a direct influence of good work-life balance over one's quality of life [3]. Greenhaus et al. provided several insights into the relation between work-family balance and the quality of life and argued that the balance does enhance an individual's quality of life, as they believe that balanced individuals experience lower levels of stress when enacting multiple roles. They argued that under certain conditions, work-life balance is associated with the quality of life; namely when individuals invest a substantial amount of time or involvement in their combined work and family roles, the degree of balance has implications for an individual's quality of life. They also confirmed the negative effect of work imbalance on quality of life and demonstrated that the deterious effects result from the raised level of conflict and stress [11]. Kofodimos has suggested that when there is imbalance, it affects the quality of life in an adverse way by arousing high level of stress [16]. Marks and MacDermid found that the people who lived balanced lives experienced, amongst others, less depression than their imbalanced counterparts [17]. Ramos's research results showed that individuals who maintain some aspects of balance, experience better quality of life [18]. Meenaksh et al. link the lack of balance to the consequences of prolonged stress, such as heart disease, smoking, alcoholism, weight gain, depression or diabetes; they also notice that "without creating a work-life balance a person is not able to take time to enjoy the life they have worked so hard to create" and takes "the stress out on the ones they love" [7]. In other words, work-life balance is an important factor that must not be neglected in pursuit of good quality of one's life.

**The definition of machine learning/ neural networks.** A natural brain has the abilities to learn new things, adapt to new and changing environments, analyse incomplete, unclear, fuzzy information and draw conclusions and judgements based on it. For example, a baby has the ability to recognize their mother from the touch, voice and smell; it is possible to identify a person from a blurry photograph, providing we know them. The brain has also the capacity to store large amounts of data. Kukreja explains that an artificial neural network (ANN) "in its simplest form (...) is an imitation of the human brain" [19]. Gurney describes ANN as "an interconnected assembly of simple processing elements, *units* or *nodes*, whose functionality is loosely based on the animal neuron" [20]. An artificial neuron network is composed of processing units. The units, called neurons, try to replicate the structure and behaviour of the natural neuron. The structure can be "trained" [19]. The processing ability of the network is stored in the interunit connection strengths, or *weights*, obtained by a process of adaptation to, or *learning* from, a set of training data [20]. The commonly used method of training multilayer ANN is the backpropagation algorithm. In it, an ANN is first given a set of known input data and asked to obtain an output, which is compared to the correct answer, the network's error is calculated, and the weights are adjusted through the backpropagation algorithm to obtain output which is closer to the known answer, as illustrated in Fig 1; this process is called training the network.

## Dataset

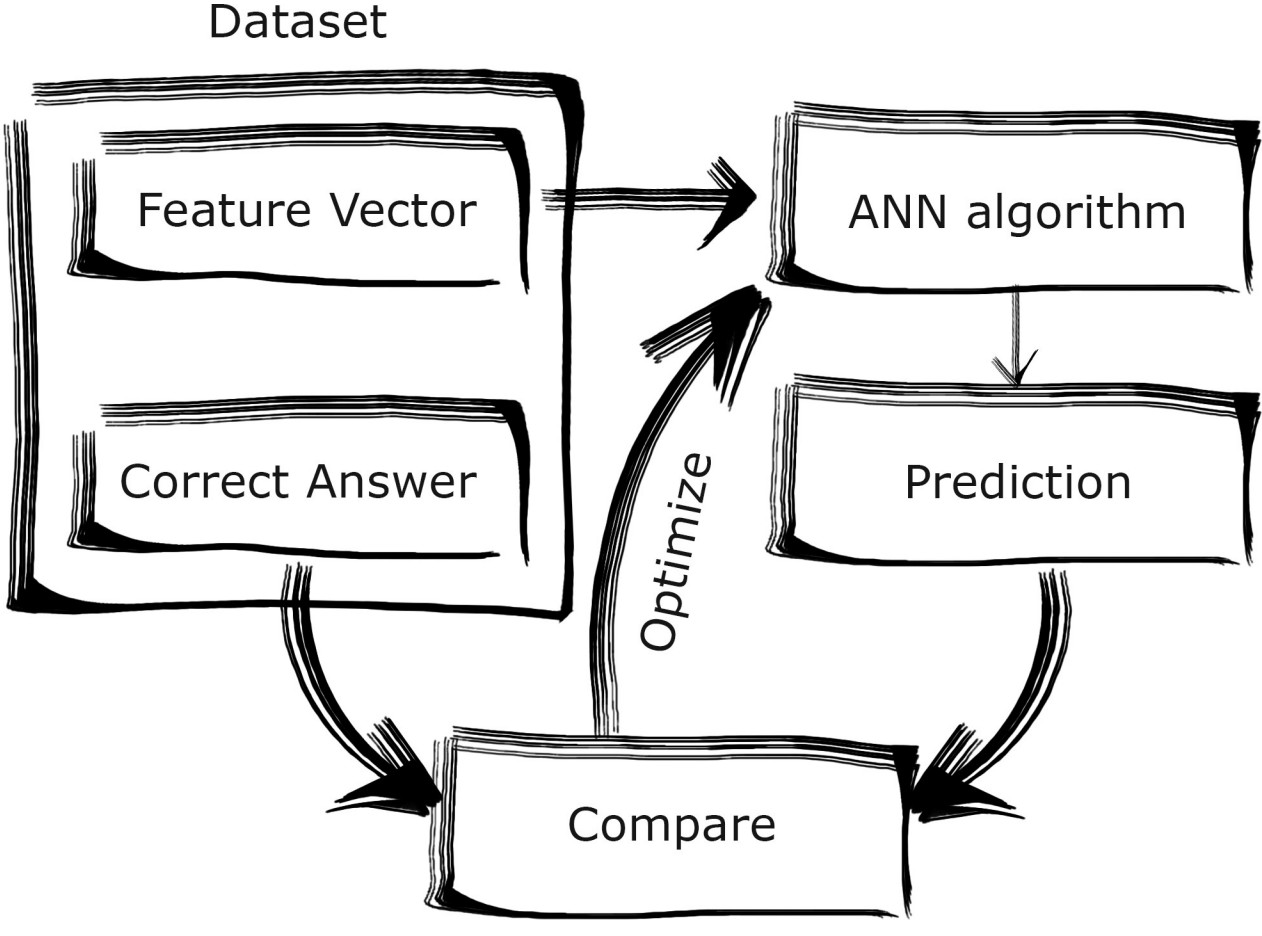

**Fig 1. Artificial network training process.**

The network undergoes many cycles like this on batches of different datapoints, until the error stops decreasing; the network is then said to be trained. Then, the trained network will be able to predict a correct output based only on the input data. This is illustrated in Fig 2. [19].

ANNs differ from normal computer programs in several ways. Firstly, they are able to learn in an adaptive way, i.e. unlike ordinary programs that follow a certain procedure designed by the coders, ANNs are able to learn how to perform a certain task solely from the presented data. This gives ANNs the adaptiveness required to perform complex tasks that rely on finding patterns in data—like face or voice recognition, intrusion detection in cybersecurity and a

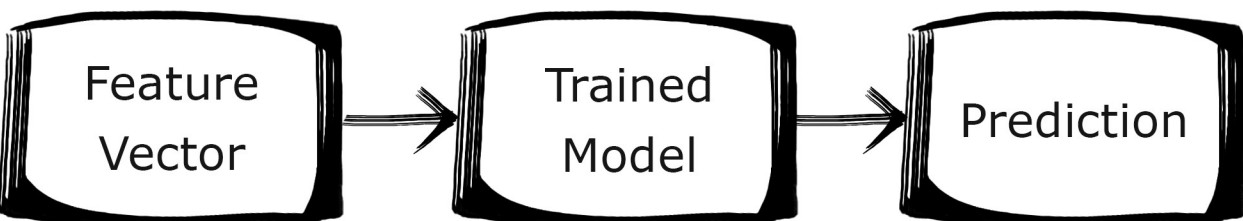

**Fig 2. The ANN-based model used to make prediction from the set of independent variables, called the feature vector.**

myriad of other complex applications. ANNs have been called a "versatile tool for modeling", "a standard utility for data mining, providing classification, regression, clustering and time series analysis abilities". ANNs prove so useful, as "not only can the network attain the relations between the variables, but it can generalize to a sufficient extent so as to allow adequate performance on unforeseen data [21].

**The relation between machine learning vs. work-life balance: A common assumption.** Undoubtedly, technology has already improved people's lives in several different ways; people can live longer and healthier lives because of technological advancements [7]. It is hoped that access to machine learning, neural network intelligence and deep learning will play a significant role in improving work-life balance in the future. This will be done by developing automation processes which in turn could lead to more efficient practices at workplace. It is speculated that machine learning can potentially improve work-life balance by reducing the amount of "grunt work", helping in organizational management, enhancing productivity, increasing workplace productivity quotient or applying hyper-personalization. Thus, automation is hoped to relieve the stress of cramping too many tasks in workers' days that in long run is detrimental to their personal relationships, health and happiness [2]. However, can machine learning be used to measure one's work-life balance, or help to find the ways of achieving it?

**State of the art.** The presented study was innovative, as up to date there have not been many attempts to employ a machine learning tool to try and predict the factors which may be related to the feeling of balance that would be recorded in the literature. In fact, the study was the first of the kind in Poland. The subject literature presents merely a few similar trials from other countries in the world. Devadoss has made an attempt to analyse the factors that induce work-life imbalance in organizational settings using the fuzzy model Induced Fuzzy Cognitive Maps. They found that the personal factor of "too much of household activities" was the first one to affect individuals' WLB. According to the authors, this factor is triggered by most of the factors, such as "unsupported spouse", "financial burden", "no support from family members" and "difficulties in caring ill/old family members, that is the impact of these factors leads to too many household activities and consequently to work-life imbalance. The other most significant factor was "health problems", triggered by "inadequate sleep", "no proper food" and "financial burden" [22]. This set of variables varies greatly from the ones used in this paper; moreover, the authors seem to have concentrated more on individual's personal life affecting the overall WLB.

Preetha et al. used an artificial neural network to determine if quality of work life and any impact on employees' mental health. However, they again chose a different set of variables to determine workers' levels of the quality of life. They included: "work implication, motivation of interior job, need of strength in higher order, realized characteristics of interior job, contentment of job, life contentment, joyfulness, self rated anxiety [23]".

Finally, Anand et al. examined if the relationships between several demographic variables, i.e. age, gender, educational qualification, marital status etc. and individual variables, such as working hours, family, co-workers, etc. influence the feeling of employee satisfaction and retention. The Chi square test they performed concluded that there was an association between workers' WLB and some demographic factors; the ANOVA test indicated that study factors did not vary with the demographic factors. Their regression model showed there was no significant effect of individual factors upon workers' WLB. It must be mentioned that the study sample consisted in 120 workers and was restricted to the workers of rural areas [24].

In [25], two global categories of variables were selected: the relations that result from the professional work influencing the personal life of an individual and the effects of the professional work influencing both the spheres of one's functioning. This led to formulating the set of detailed variables for the first global category, comprising of one's attitude towards work,

the aspects of work sought after by the workers and provided (or not) by employers, the number of hours devoted to work, the time spent on commuting, "bringing work duties home", working extra hours or additional employment, being engaged in one's work, the setup of life activities and being satisfied with them, the preferred setup of one's life activities, the significance of WLB for the particular worker and one's subjective assessment of one's WLB. The detailed variables for the second global category were: the time for themselves a person had at their disposal, the difficulties in combining work and life, interpersonal relations in one's personal life, health state, the motivation to work at the current workplace, being overburdened with work, one's assessment of their productivity, interpersonal relations at one's workplace, being ready to change jobs in order to achieve balance, as well as the solutions that would enhance the WLB of the workers [25]. The aforementioned global variables, as well as the sets of detailed variables were used as the basis for the studies and experiments described in this paper.

## Experimental setup and dataset description

The aim of the trial was to answer the research question, that is to check whether a machine learning tool is able to analyse the data given by workers and find if there exist any correlations between the various employee-specific and workplace factors and employers' subjective feeling of maintaining work-life balance.

**Dataset description.** In order to conduct this experiment, the data from an original empirical study conducted in 2017 was used. (Full description of the particular study and the detailed data are to be found in [25] and [5]). What made it unique was the fact that it was one of the first studies of this kind. Furthermore, a sample group of 800 workers had been analysed in total, making it one of the largest analyses of this type, in both Poland and other countries of the world. Taking part in the study was fully consensual and voluntary. The data obtained from the workers was anonymised, making it impossible to recognize the particular employees. Thus, the Ethics Committee of the Kazimierz Wielki University approved the gathering the data and conducting all the subsequent studies on the resulting dataset.

The study was of nationwide range; it was carried out in 80 randomly chosen organizations; representing the 16 sections of economy according to the Polish Classification of Business Activities (PKD), (5 companies for each branch). In each of the companies, 10 workers were requested to fill in the study questionnaire. The workers were drawn from the employees' register provided by the employers by the investigators at random.

Some of the characteristics of the surveyed employees are presented in Tables 1–4:

The study aimed at identifying the relationships that result from the influence of work over the life of an individual. Then, the identified relations were to be described and their nature—explained. Moreover, the researchers wished to examine the possible effects of that influence over both the workers' professional work and personal lives.

Some other aims of the study were, e.g. to find if workers in Poland think they maintain the balance between their professional work and private lives, and if there is any relationship between the feeling of balance and the branch of economy the employees work at; the study produced some very promising results [5].

**Actual working time vs. WLB.** The respondents were divided into three groups, according to the proportion of their contracted working time and the actual working hours. The study results show that:

**Table 1. Gender of the respondents (N = 800).**

| Men | Women |
|---|---|
| 389 | 411 |

**Table 2. Age of the respondents (N = 800).**

| Respondent's age | 18–29 years | 30–39 years | 40–49 years | 50–59 years | 60 and over |
|---|---|---|---|---|---|
| Number of people | 101 | 286 | 245 | 145 | 23 |

**Table 3. Respondents' job experience (N = 80).**

| General job experience in years | 1–5 | 6–10 | 11–20 | 21–30 | 31–40 | Over 40 |
|---|---|---|---|---|---|---|
| Number of people | 78 | 170 | 313 | 134 | 104 | 1 |

**Table 4. The type of the enterprise the employees work for (N = 80).**

| Public institutions and companies | Private companies |
|---|---|
| 41 | 39 |

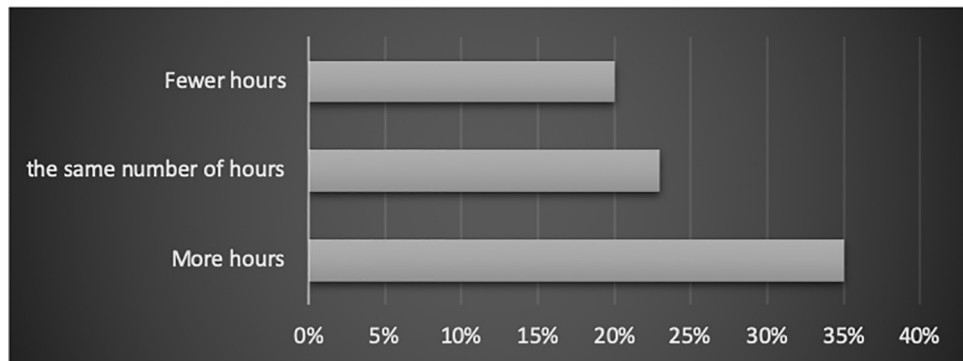

**Fig 3. The relation between the actual number of hours vs. the contracted number of hours, and employees' work-life balance (% of people in the category).**

- Among the people who work fewer hours than contracted, 20% experience some disruptions of work-life balance.

- Of the employees who work exactly the same number of hours as contracted, 23% feel they do not have full work-life balance.

- The people who work longer than contracted experience imbalance between work and personal lives the most often (35% of respondents). The relation has been shown in Fig 3.

**Free time for themselves.** The studied employees were asked how much time they have exclusively for themselves. It turned out that the level of perceived work-life balance changed according to the number of hours the person had at their disposal every week.

Among the people who had:

- 10 or fewer hours of the time for themselves a week, 40% experienced the imbalance.

- 11–20 a week– 37%

- 21–30 hours– 23%

- 31 and more hours– 10%

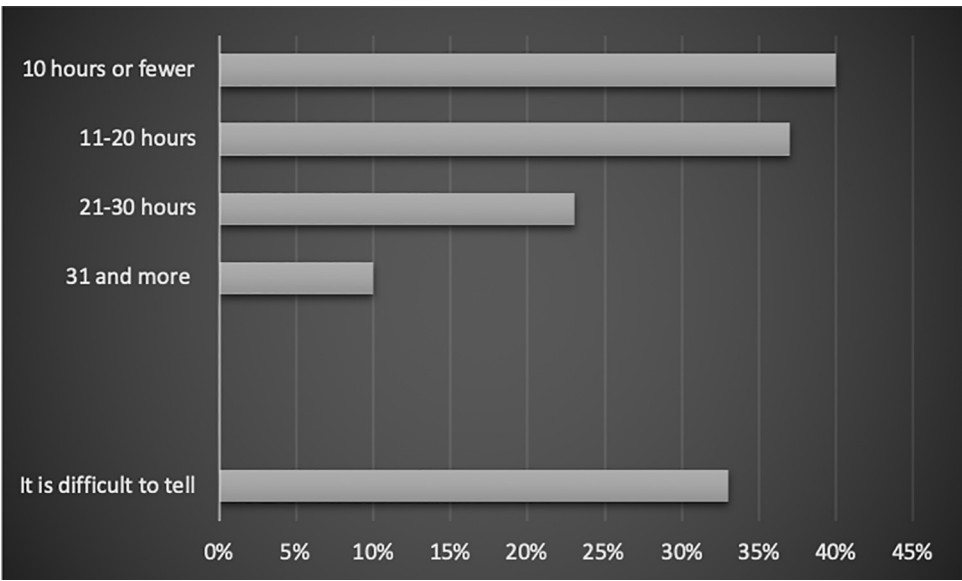

**Fig 4. The relation between the time one has at their disposal and employees' work-life balance (% of people in the category).**

There was also the answer of "It is difficult to tell", and 33% of the people who chose this answer experienced some kind of work-life balance disruptions. The relation has been shown in Fig 4.

**Working at weekends only.** In the study, work at weekends only seemed to concur with the feeling of work-personal life imbalance.

The 50.3% of employees who work only at the weekends felt their work and lives were in conflict. On the other hand, amongst the people who worked on weekdays, weekdays and weekends, or weekdays, weekends and holidays, 25% stated they felt their work or lives influenced the other in an adverse way. The relation has been shown in Fig 5.

**Being self-employed.** Seventy per cent of the people who were self-employed (ran a one-person business) experienced the conflict. In comparison, the people who were not self-employed indicated the occurrence of some form of imbalance in 28% of cases. The relation has been shown in Fig 6.

**Subjective assessment of the employees' financial situation.** Some correlation has also been found between the way the employees see their financial status and work-life balance.

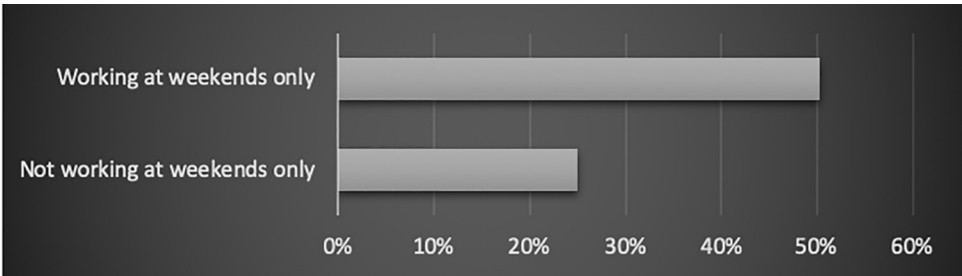

**Fig 5. The relation between working at weekends only and not doing so, and employees' work-life balance (% of people in the category).**

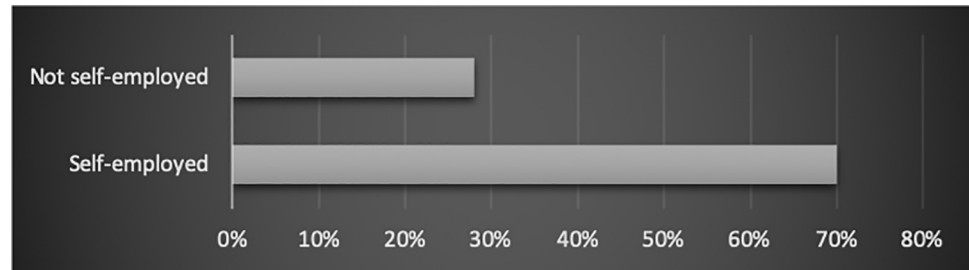

**Fig 6. The relation between being self-employed vs. not being self-employed and employees' work-life balance (% of people in the category).**

The people who found their financial situation as very good, experienced the imbalance in 24% of cases.

Good– 32%. Rather good– 20%. Neither good nor bad– 36%. Rather bad– 33%. Bad– 53% and Very bad– 50%.

The relation has been shown in Fig 7.

The data the network analysed comprised of the employee's:

- Gender

- Age

- Education

- Marital status

- The number of children (including underage children)

- The information if the employee takes care of an adult dependant

- The subjective assessment of material situation

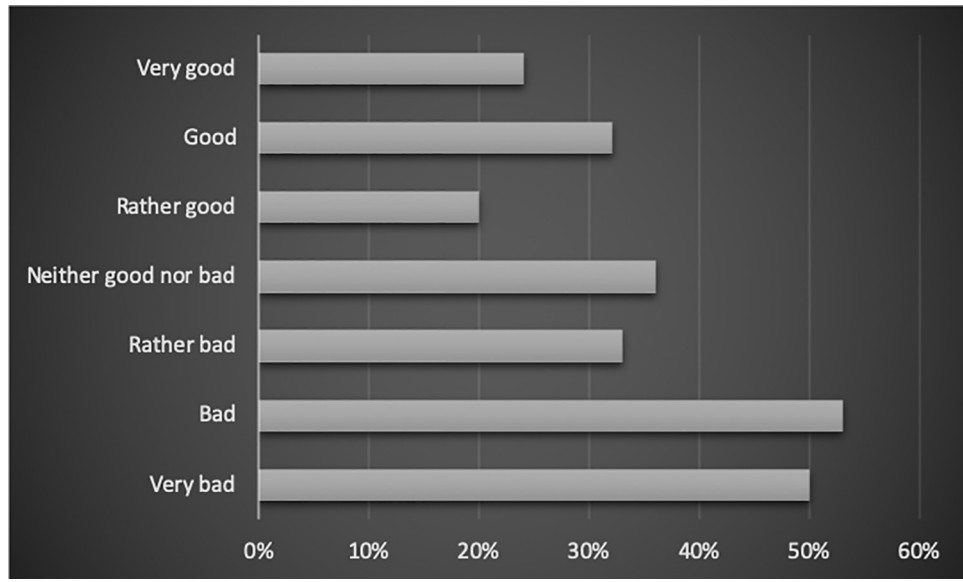

**Fig 7. The relation between the subjective assessment of one's financial situation and employees' work-life balance (% of people in the category).**

- General seniority

- Years worked in the current profession

- Years worked at the current workplace

- The size of the organization

- Occupational category

- The type of job contract (fixed-term contract or permanent employment; full-time part-time or contract, a specific-task contract, contract of mandate, no contract, self-employed, own company, farmer).

- The contracted working hours a week

- The actual working hours

- Working on weekdays only; weekdays and weekends; weekends only; weekdays, weekends and holidays

- Working only in the mornings, afternoons or at night or working different times

- The hours devoted to commuting every day

- The number of extra hours

- Any additional employment

- The number of hours a week of time for themselves

- And if their workplace applies any solutions towards WLB.

All the above-mentioned data was compared with the workers' answers to the question: *What kind of difficulties do you experience when combining your work with personal life?* The possible answers were:

1. work affects my personal life in an adverse way,

2. my personal life affects my work in an adverse way,

3. both of the above,

4. neither of the above.

For the sake of this study, it has been assumed that the answers 1–3 mean there is some disruption in employee's work-life balance, whilst the answer 4 means the balance is maintained.

**Architecture of the machine learning tool.** The Artificial Neural Network used in this experiment was designed with limited data in mind. Thus, with input vector reduced to 10 features, the ANN of 2 hidden layers and a softmax layer was compiled. The first hidden layer received 50 neurons, the second 25 neurons, both used the Rectified Linear Unit (ReLU) activation function. The used optimizer–the part of the algorithm responsible for updating the weight parameters to minimize the loss function—of the ANN was adaptive moment estimation (ADAM), while the loss function was Categorical Crossentropy. The batch size and the number of epochs were empirically found through a number of iterations to obtain the highest accuracy. The values for those were batch size: 20 and epochs: 250. This particular setup emerged after performing a number of tests, which revealed that architectures with smaller number of neurons did not perform as well as the chosen one, and architectures larger did not result in getting better results. This was true for the number of hidden layers as well. The ANN

algorithm has a number of parameters that can be chosen by the designer. These are called hyperparameters. The hyperparameter setup was established with the grid search method. This method does an exhaustive search over the hyperparameter space. Different activations functions, optimizers, batch sizes, number of epochs and different number of neurons were tested.

**Data preprocessing and the experimental pipeline.** A number of features in the dataset were of the categorical type. For the machine learning method to be able to process this kind of data it has to be encoded as separate columns with 1 signifying the presence of a category in a particular datapoint and 0 the absence of it. Additionally, the original dataset contained fields where the respondents did not supply any answer. These datapoints were removed from the dataset entirely. As per domain standard, the dataset is split into two parts: one used to train the algorithm and the other used to test it. Since the algorithms need a lot of data samples to converge, the split is not usually uniform. The dataset in this study was then split into the parts used for training and for testing the ML tool using the ratio of 9:1. The experimental pipeline is illustrated in Fig 8. For the network to obtain optimal results the input data need to be pre-processed. In the preprocessing step 10 most influential features were selected using the SelectKBest sklearn.feature_selection algorithm [26]. The algorithm measures the correlation

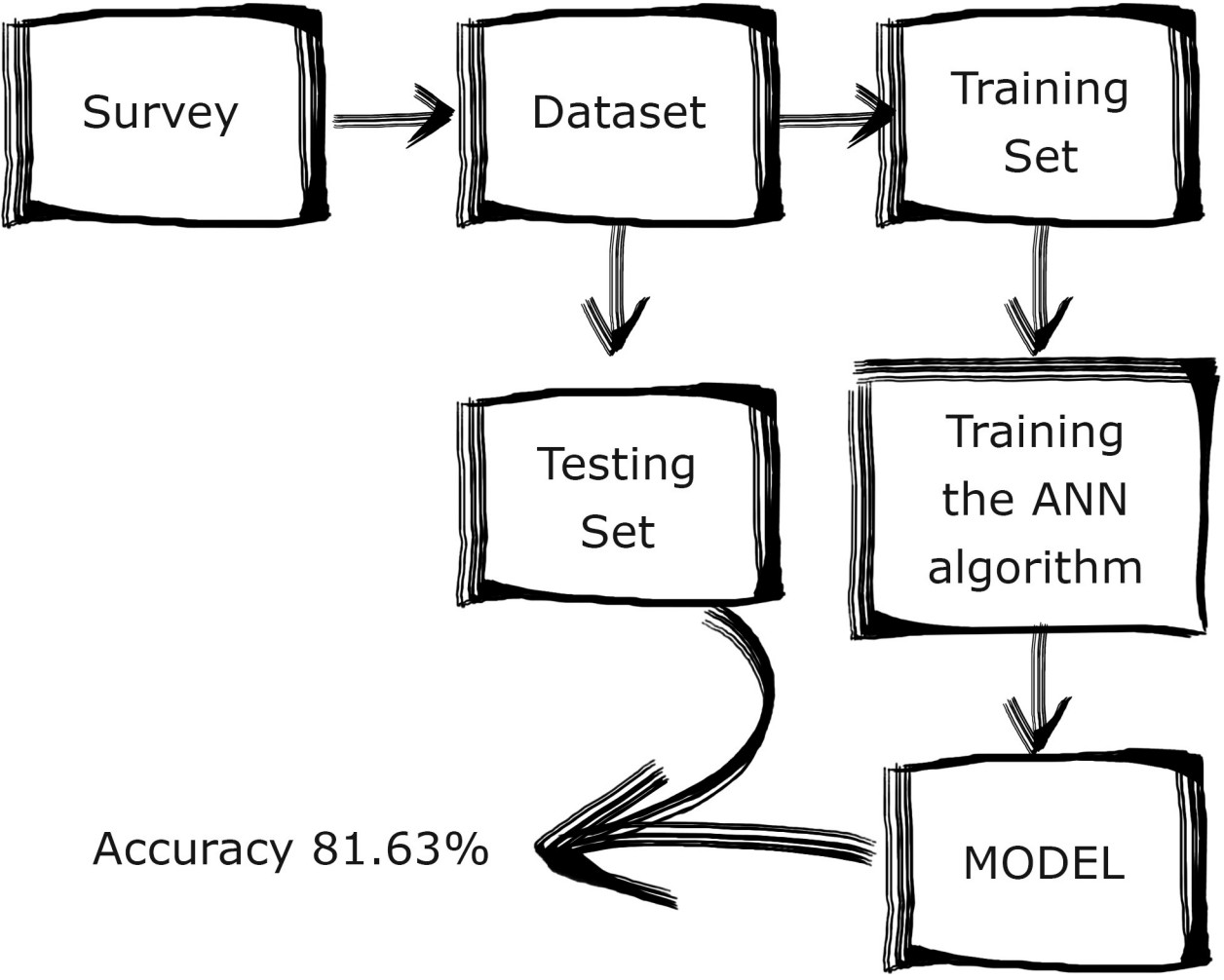

**Fig 8. The experimental pipeline.**

between the independent variables and the dependent variable using one of the provided metrics, chi$^2$ in this example. The algorithm indicated that there was a sharp decline in the correlation after the $10^{th}$ feature. Those features are:

- Contracted working time

- Actual working time

- Working weekends only

- Free time

- Financial status

- Years worked at the current job

- Self-employed

- Economy sector

- The size of the company

The cleaned dataset had 800 usable observations, with 244 cases reporting the lack of WLB and 556 reporting WLB as satisfactory. This shows imbalance among the two classes in the dataset, so the majority class was randomly subsampled down to 239 samples to achieve a better balance of classes.

## Results

### The answer to the research question no. 1

Having used the proposed innovative methodology and ML based tool we were able to establish that the ANN is capable of finding the correlations existing in the dataset. Using the answers to the survey questions it is able to predict whether a person reported balance or the lack thereof with 81.63% accuracy. The detailed results are displayed in Table 5.

This answered the first research question. The most significant was the relation between the actual working hours. Amongst the other noteworthy factors, there were:

- the amount of free time a week the employee has for themselves,

- working at weekends only,

- being self-employed,

- subjective assessment of one's financial status.

### The answer to the research question no. 2

In order to find the answer to the question of how shifting the parameters would probably influence the workers' subjective feeling of WLB, classification experiments had to be

**Table 5. Classification report.**

|           | precision | recall | f1- score | support |
|-----------|-----------|--------|-----------|---------|
| **0**     | 0.88      | 0.64   | 0.74      | 22      |
| **1**     | 0.76      | 0.93   | 0.83      | 27      |
| **micro avg** | 0.80  | 0.80   | 0.80      | 49      |

performed. For the use of this experiment, an artificial neural network was constructed. Indeed, interfering in certain values had considerable influence over the changes in the perceived WLB.

**The classification experiments.** The network of the above-mentioned architecture was trained on the described dataset to predict whether the person is likely to report work-life imbalance, or not, basing on the 10 independent variables selected as best predictors in the dataset. The ANN achieved 81.63% accuracy on the test set. Accuracy is counted as the number of correct predictions divided by the number of total predictions.

From the testing set a number of correctly classified examples were taken out to perform the experiments in the following section. In the experiments a specific feature of those samples was altered to see whether (or at what point) would it cause the ML tool to switch classification from one to the other.

*Experiment 1.1.* For this experiment, a person who worked 50 hours a week (contracted:40) and did not experience work-life imbalance was chosen. The parameter of the actual working time was changed; from 10 to 150 hours a week. The rest of parameters remained unchanged. The network predicted that there would be no conflict if the person worked 10, 20 or 30 hours a day. At 40 hours it predicted there would be some imbalance, at 50 –the lack thereof. For an employee working from 60 to 150 hours a week, the network always predicted some imbalance.

The skip between classes at about 50 hours may suggest that this is the decision boundary. As the network's accuracy is slightly below 82%, there may be no pinpoint accuracy.

*Experiment 1.2.* Then, a person working 37 hours a week (contracted: 37) with no imbalance was chosen. This particular sample was of interest because of the false positive–the ANN classified the sample as a person reporting imbalance, but the person themselves reported no imbalance. The working time was gradually changed, from 5 to 85 hours, adding 5 hours each time. The network predicted that if the working time were shorter than 35 hours a week, there would be no imbalance. Some conflict would occur if the working time were longer than 40 hours a week.

The course of the changes in the subjective WLB predicted by the ANN has been presented in Table 6.

*Experiment 1.3.* A person working 100 hours (out of 40 contracted), who claimed they experienced imbalance. The number of working hours a week was gradually lowered from 100 to 5, subtracting 5 hours a time. The network predicted that between 30 and 25 hours of work a week, balance would be found.

Then, the subjective assessment of one's material situation was scrutinized.

*Experiment 1.4.* Firstly, a person who claimed their material situation was "very good" and achieved balance was chosen. Then, the financial situation of the person was gradually changed from "very good" to "very bad", but the network predicted it would not affect their work-life balance. Next, their working hours were increased. According to the prediction, some imbalance would occur at 75 hours of work a week.

*Experiment 1.5.* On the other hand, a person who worked 40 hours a week (contracted: 40), found their financial situation very bad and experienced some imbalance was selected. The network did not predict any changes in the feeling of balance even when the financial situation improved, up to the level of "very good". Then, the working hours were lowered; the network predicted that the worker would finally achieve balance when the number of working hours a week amounted to 10.

Next, the time for oneself was looked upon.

*Experiment 1.6.* A person who reported they had more than 31 hours of time a week for themselves, worked 38 hours a week (contracted: 38) and did achieve balance was selected.

**Table 6. The influence of the simulated shifts in working time on subjective WLB according to the ANN.**

| Contracted working time | Actual working time | Working weekends only | Free time | Financial status | Years worked at the current job | Self-employed | Economy sector | The size of the company | Subjective WLB |
|---|---|---|---|---|---|---|---|---|---|
| 40 | 5 | no | Hard to tell | Neither good nor bad | 5 years | no | 1 | 10–49 people | balance |
| 40 | 10 | no | Hard to tell | Neither good nor bad | 5 years | no | 1 | 10–49 people | balance |
| 40 | 15 | no | Hard to tell | Neither good nor bad | 5 years | no | 1 | 10–49 people | balance |
| 40 | 20 | no | Hard to tell | Neither good nor bad | 5 years | no | 1 | 10–49 people | balance |
| 40 | 25 | no | Hard to tell | Neither good nor bad | 5 years | no | 1 | 10–49 people | balance |
| 40 | 30 | no | Hard to tell | Neither good nor bad | 5 years | no | 1 | 10–49 people | balance |
| 40 | 35 | no | Hard to tell | Neither good nor bad | 5 years | no | 1 | 10–49 people | balance |
| 40 | 40 | no | Hard to tell | Neither good nor bad | 5 years | no | 1 | 10–49 people | balance |
| 40 | 45 | no | Hard to tell | Neither good nor bad | 5 years | no | 1 | 10–49 people | No balance |
| 40 | 50 | no | Hard to tell | Neither good nor bad | 5 years | no | 1 | 10–49 people | No balance |
| 40 | 55 | no | Hard to tell | Neither good nor bad | 5 years | no | 1 | 10–49 people | No balance |
| 40 | 60 | no | Hard to tell | Neither good nor bad | 5 years | no | 1 | 10–49 people | No balance |
| 40 | 65 | no | Hard to tell | Neither good nor bad | 5 years | no | 1 | 10–49 people | No balance |
| 40 | 70 | no | Hard to tell | Neither good nor bad | 5 years | no | 1 | 10–49 people | No balance |
| 40 | 75 | no | Hard to tell | Neither good nor bad | 5 years | no | 1 | 10–49 people | No balance |
| 40 | 80 | no | Hard to tell | Neither good nor bad | 5 years | no | 1 | 10–49 people | No balance |
| 40 | 85 | no | Hard to tell | Neither good nor bad | 5 years | no | 1 | 10–49 people | No balance |

Then, their time for themselves was lowered but the network did not predict any conflict. The actual number of working hours was then tested, from 5 to 78 hours a week. As the network predicted no imbalance, it was raised to 168 hours a week. It has to be noted that the ANN treats the number of hours as an integer like any other continuous value and will allow to raise the number indefinitely. However, the values found in the dataset are real-world values, therefore picking an unreasonable number of working hours is an interesting exercise, but highly unlikely to produce results that fall outside of the ones expected to be found in the dataset. The network predicted there would be no imbalance as long as the person had more than 31 hours a week at their own disposal.

*Experiment 1.7.* An employee working 60 hours a week (contracted: 40), with fewer than 10 hours a week of the time for themselves, reporting the imbalance was selected. Giving the person more time for themselves would not result in creating balance, according to the network's prediction. However, when the working hours were gradually lowered, down to 25 hours week, the network predicted the imbalance would disappear if the person worked just 10 hours less (50).

## Conclusions

Although work-life balance has been the subject of a widespread public debate, it has been widely accepted that workers do need to enjoy reasonable balance between their work and personal lives. Furthermore, a number of benefits have been attributed to the maintaining of WLB, including better performance, productivity and competitiveness at work and raised morale and motivation. At the same time, it is believed to lower the levels of stress, sickness and absenteeism. In many countries the actions aimed at fostering better work-life balance and supporting working families have become crucial part of government policies.

The trial with the use of the ML tool indicated that there exist some factors the presence or lack of which may influence the workers' perceived work-life balance–which translates into their quality of life, too. This could be of use, as knowing the factors that may be conducive to the feeling of having better work-life balance will allow to predict possible areas of intervention; this may be especially crucial for employees and organizations. For example, being aware of the fact that the number of hours worked a week is related to work-life balance, employers or organizations may wish to apply work-life balance solutions to the workers who work longer hours than they are supposed to according to their work contracts. Furthermore, the ML tool is capable of specifying, in some cases, the estimated point where a subject's decision boundary resides, and if this boundary is crossed the WLB classification becomes inverted. This is a very important piece of information for both the employer and the employee; therefore the approach is validated.

It is possible that some workers are not aware of the ways the relations among various factors influence their personal balance. As the above-mentioned network could find dozens of other correlations and conduct far more experiments, it may potentially be utilized to find the sweet spots for the given factors for individual employees. Then, after it helped to identify the factors that should be changed and indicated the adjustments that could be made in the employees' lives, it might help workers enjoy the balance they longed for.

Indeed, the experiments helped to find the particular factors that might influence the workers' subjective feeling of WLB. It has been found that the most significant was the relation between the feeling of balance and the actual working hours. In the experiment, raising the number of working hours for a person with perfect balance resulted in the appearance of imbalance at one point. The ANN predicted that similarly, for a person working long hours some balance would be found when the actual number of hours were lowered. Amongst the other noteworthy factors, there were:

- the amount of free time a week the employee has for themselves,

- working at weekends only,

- being self-employed,

- subjective assessment of one's financial status.

The sample group of 800 employees analysed for the sake of the study makes the results relevant and valid. Moreover, the significant width and breadth of the study, along with the innovative use of an emerging way to extract meaningful information from data, i.e. artificial neural network, for analysing the data could become a new direction for the course of future studies. In this work, the specific setup of the tool, along with the description of the dataset have been provided in order to ensure replicability of the study. Nevertheless, further future research is necessary to check whether other machine learning tools will find similar correlations, and if the correlations found in the sample will be reflected in other groups of workers.

## Author Contributions

**Conceptualization:** Aleksandra Pawlicka.

**Data curation:** Marek Pawlicki, Renata Tomaszewska.

**Formal analysis:** Michał Choraś.

**Investigation:** Renata Tomaszewska.

**Methodology:** Aleksandra Pawlicka, Renata Tomaszewska.

**Project administration:** Marek Pawlicki.

**Resources:** Ryszard Gerlach.

**Software:** Marek Pawlicki.

**Supervision:** Michał Choraś, Ryszard Gerlach.

**Validation:** Michał Choraś.

**Writing – original draft:** Aleksandra Pawlicka.

**Writing – review & editing:** Marek Pawlicki, Michał Choraś.

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
