## [Decision Letter · Decision Letter 0]

19 Mar 2020

PONE-D-20-02847

Innovative artificial intelligence approach and evaluation campaign for predicting the subjective feeling of work-life balance among employees

PLOS ONE

Dear Dr Pawlicka,

Thank you for submitting your manuscript to PLOS ONE. After careful consideration, we feel that it has merit but does not fully meet PLOS ONE’s publication criteria as it currently stands. Therefore, we invite you to submit a revised version of the manuscript that addresses the points raised during the review process.

Your manuscript has been assessed by two reviewers, as you will see in the comments provided below. Overall, the valuation of both experts has been positive and, added to my personal opinion on it, I believe it is suitable for being published in PLOS ONE. Nevertheless, it requires a set of amendments and modifications (most of them minor), that you should perform through a revision of the paper along the next weeks. Some key descriptions and clarifications and further implications of your work are suggestible to be included in this revised version of the manuscript. Please notice that the comments of the Reviewer # 2 are embedded to a sanitized (commented) document that is available next to their review report.

We would appreciate receiving your revised manuscript by May 03 2020 11:59PM. To enhance the reproducibility of your results, we recommend that if applicable you deposit your laboratory protocols in protocols.io, where a protocol can be assigned its own identifier (DOI) such that it can be cited independently in the future. For instructions see: http://journals.plos.org/plosone/s/submission-guidelines#loc-laboratory-protocols

We look forward to receiving your revised manuscript.

Kind regards,

Sergio A. Useche, Ph.D.

Academic Editor

PLOS ONE

Journal Requirements:

2. Please amend your Competing interests statement to declare any competing interests such as author commercial affiliations, or state that there are no competing interests.

3. Thank you for including your ethics statement:

"N/A

Although the study involved human participants, the study was completely anonymous and fully consensual. Thus, the Ethics Committee of the Kazimierz Wielki University fully agreed to perform all the experiments. "

Please amend your current ethics statement to confirm that your named institutional review board or ethics committee specifically approved this study.

4. We suggest you thoroughly copyedit your manuscript for language usage, spelling, and grammar. If you do not know anyone who can help you do this, you may wish to consider employing a professional scientific editing service.  

5.****We note that you have indicated that data from this study are available upon request. PLOS only allows data to be available upon request if there are legal or ethical restrictions on sharing data publicly. For more information on unacceptable data access restrictions, please see http://journals.plos.org/plosone/s/data-availability#loc-unacceptable-data-access-restrictions.

6. Thank you for stating the following in the Competing Interests section:

We note that one or more of the authors are employed by a commercial company: ITTI, Bydgoszcz.

7. Please ensure that you refer to Figure 3-7 in your text as, if accepted, production will need this reference to link the reader to the figure.

8. We note you have included a table to which you do not refer in the text of your manuscript. Please ensure that you refer to Table 5 in your text; if accepted, production will need this reference to link the reader to the Table.

Reviewers' comments:

Reviewer's Responses to Questions

**Comments to the Author**

1. Is the manuscript technically sound, and do the data support the conclusions?

Reviewer #1: Yes

Reviewer #2: Yes

2. Has the statistical analysis been performed appropriately and rigorously? 

Reviewer #1: N/A

Reviewer #2: Yes

3. Have the authors made all data underlying the findings in their manuscript fully available?

Reviewer #1: Yes

Reviewer #2: No

4. Is the manuscript presented in an intelligible fashion and written in standard English?

Reviewer #1: Yes

Reviewer #2: Yes

5. Review Comments to the Author

Reviewer #1: Dear authors,

I want to congratulate you for your present work and interest in publication. It will be a valuable source of knowledge for interested researchers on the field.

In my professional opinion, your text is relevant and suitable but needs a few improvements to achieve the expected quality for the journal. I will go through them in order of appearance in your manuscript:

[Abstract]

Dear authors. Please give more insights about the methodology, models and the results obtained on your abstract, this section must summarize the project as possible, Have in mind that could be the only part of your work that could be read in a quick search for external researchers.

[Introduction]

Artificial intelligence is a wide knowledge field. I strongly suggest to narrow your terms and use “machine learning” or “deep learning” instead. This should be applied even to the title of the article.

Line 57: when you mention the group of 800 employees as “exceptionally large”, which are your references for this comparison? Maybe other studies on similar cases? The sampling of the population on the determined geographical area? Please clarify.

Also, at this point will be useful for the readers to make clear about the geographical and demographic limitations of your research. Please clarify.

[Materials and Methods]

Line 121: Please make a further description on the three chosen variables for the experiments. As they are written “time balance”, “involvement balance” and “satisfaction balance” are quite ambiguous and hard to measure for the readers.

After reading the section is clear that there is no consensus on the definition or how to measure the WLB. So, it’s important to specify which of the exposed will be the definition used specifically for this research. Or, if you’re introducing your own definition, please make it clear.

[State of the art]

As show in the State of the art, the innovation value of your research seems to be only comparable with projects on Poland. Is the state of the art limited for the country or aren’t examples available outside it?

In the state of the art you review models based on Fuzzy Cognitive maps and ANN for similar research topics. This reinforces my idea that you should be clearer on the real innovative value of the research. Please try to describe the major differences and improvements in comparison of these projects.

[Dataset description]

Link to reference (25) is broken. Please find an alternate one.

Line 237: After reading the referenced paper on (5) I assume that the 16 branches of the economy are the “16 sections of the Polish Classification of Business Activities” but as a foreign reader it was difficult to me to realize that. Please make it clear on this text.

[Architecture of the AI tool]

What was your criteria for selecting that architecture for the ANN?

Did you make other experiments with alternative configurations?

Even though the results were favorable, I’m concerned about the model optimization process. The architecture appears to be an arbitrary choice. That’s not necessarily wrong, but it will be better if you explain your decision.

[Data preprocessing and the experimental pipeline]

In order to improve the readability and structure of the manuscript, I recommend you to move the details of the feature selection techniques used on the experiments from the line 390 to 394 to this section. Please give a further description of the 10 selected features and try to justify the usage of the SelectKBest algorithm over other alternatives.

[Results]

Was your training dataset balanced? I couldn’t find anywhere the ratio of observations for each of the 4 categories. This could be decisive when calculating the accuracy of the predictions. Please clarify this or attach the classification report including other metrics like f1, precision and recall, ROC AUC, etc. I’m afraid that under this circumstances accuracy won’t be enough to answer the Research Question 1.

The selected method for

[Conclusions]

I would like to read here your conclusions and personal appreciations about the work done using ANN models. Do you conclude it was a good approach? Could you compare with other possibilities?

Please, be specific here on the findings on which were the more meaningful variables for the prediction. This information could be extracted from the results of the feature selection algorithm.

Don’t hesitate to tell us detailed technical conclusions of your work. It may be important for the researchers to know about the troubles you had to face, your thoughts and considerations over the data extraction and preprocessing, the used model and the validation and evaluation of the results.

Reviewer #2: This is a well written paper and an innovative approach. Please see my comments on the attached file. I strongly encourage the changes that i recommended be implemented before acceptance for publication

6. PLOS authors have the option to publish the peer review history of their article (what does this mean?). If published, this will include your full peer review and any attached files.

Reviewer #1: Yes: Ariel Ortiz Beltrán

Reviewer #2: No

---

## [Author Response · Author response to Decision Letter 0]

25 Mar 2020

Dear Reviewers and Editor,

We are very thankful for reading our paper and giving so much valuable feedback. It made us really happy that you found our work interesting. We would like to thank you for your positive approach towards our paper. In this revision, we have implemented all the suggested alterations and we hope that the paper can now be accepted for publication in Plos ONE. 

In particular, we have reacted to the reviews and implemented the following changes:

Editor’s suggestions:

We have implemented the suggested changes wherever applicable. The data used in this study, as suggested, has been placed in openly available repository. We have added the missing references to tables and figures. 

Reviewer #1’s suggestions:

I want to congratulate you for your present work and interest in publication. It will be a valuable source of knowledge for interested researchers on the field.

In my professional opinion, your text is relevant and suitable but needs a few improvements to achieve the expected quality for the journal. I will go through them in order of appearance in your manuscript:

Thank you so much for your kind, heartwarming words! 

[Abstract]

Dear authors. Please give more insights about the methodology, models and the results obtained on your abstract, this section must summarize the project as possible, Have in mind that could be the only part of your work that could be read in a quick search for external researchers.

We have extended description of the methodology and the relevant summary for the method was added. We have also given more details on the results in the abstract. (pp. 2 -3)

[Introduction]

Artificial intelligence is a wide knowledge field. I strongly suggest to narrow your terms and use “machine learning” or “deep learning” instead. This should be applied even to the title of the article.

Indeed, we agree with the Reviewer – many thanks for this comment.

As suggested, we have narrowed down the term to “machine learning”. Although in the case of our study it could have been used interchangeably with the term “deep learning”, we used the former term in order to make it clearer for more readers.

Line 57: when you mention the group of 800 employees as “exceptionally large”, which are your references for this comparison? Maybe other studies on similar cases? The sampling of the population on the determined geographical area? Please clarify.

Also, at this point will be useful for the readers to make clear about the geographical and demographic limitations of your research. Please clarify.

Those points have been clarified in the revised version (Introduction, page 4). 

[Materials and Methods]

Line 121: Please make a further description on the three chosen variables for the experiments. As they are written “time balance”, “involvement balance” and “satisfaction balance” are quite ambiguous and hard to measure for the readers.

After reading the section is clear that there is no consensus on the definition or how to measure the WLB. So, it’s important to specify which of the exposed will be the definition used specifically for this research. Or, if you’re introducing your own definition, please make it clear.

The three kinds of balance referred to one of the approaches found in the literature, not our own approach. We have clarified that and made the fact that we use our own definition stand out more. 

As show in the State of the art, the innovation value of your research seems to be only comparable with projects on Poland. Is the state of the art limited for the country or aren’t examples available outside it?

We have clarified this matter by adding more information (State of the Art, pp. 10-11 as well as Introduction, p. 4).

In the state of the art you review models based on Fuzzy Cognitive maps and ANN for similar research topics. This reinforces my idea that you should be clearer on the real innovative value of the research. Please try to describe the major differences and improvements in comparison of these projects.

Thank you so much for pointing it out! We have rebuilt and enhanced the whole State of the Art section so that it explains in which way the other studies differed and why our study was innovative.

Link to reference (25) is broken. Please find an alternate one.

It has been checked and should be working now.

Line 237: After reading the referenced paper on (5) I assume that the 16 branches of the economy are the “16 sections of the Polish Classification of Business Activities” but as a foreign reader it was difficult to me to realize that. Please make it clear on this text.

We have made it clearer, by adding more details in the section “Dataset description”, p. 13. Thank you for pointing this out. 

[Architecture of the AI tool]

What was your criteria for selecting that architecture for the ANN?

Did you make other experiments with alternative configurations?

Even though the results were favorable, I’m concerned about the model optimization process. The architecture appears to be an arbitrary choice. That’s not necessarily wrong, but it will be better if you explain your decision.

We have included an explanation of the choice of architecture in the text (Page 19, The section “Architecture of the Machine Learning tool” and onwards). 

[Data preprocessing and the experimental pipeline]

In order to improve the readability and structure of the manuscript, I recommend you to move the details of the feature selection techniques used on the experiments from the line 390 to 394 to this section. Please give a further description of the 10 selected features and try to justify the usage of the SelectKBest algorithm over other alternatives.

We have moved the section to the suggested place. We have also enumerated the selected features.

[Results]

Was your training dataset balanced? I couldn’t find anywhere the ratio of observations for each of the 4 categories. This could be decisive when calculating the accuracy of the predictions. Please clarify this or attach the classification report including other metrics like f1, precision and recall, ROC AUC, etc. I’m afraid that under this circumstances accuracy won’t be enough to answer the Research Question 1.

The selected method for

More details of the balancing process were included in the section, along with the respective numbers of class instances before and after balancing. A classification report table was also added.

[Conclusions]

I would like to read here your conclusions and personal appreciations about the work done using ANN models. Do you conclude it was a good approach? Could you compare with other possibilities?

Please, be specific here on the findings on which were the more meaningful variables for the prediction. This information could be extracted from the results of the feature selection algorithm.

Don’t hesitate to tell us detailed technical conclusions of your work. It may be important for the researchers to know about the troubles you had to face, your thoughts and considerations over the data extraction and preprocessing, the used model and the validation and evaluation of the results.

Many thanks for this comment – indeed more technical details are placed now in the text. In particular, the conclusions on the use of ANN were added.

Reviewer #2: This is a well written paper and an innovative approach. Please see my comments on the attached file. I strongly encourage the changes that i recommended be implemented before acceptance for publication.

This is a very interesting and innovative manuscript that merges AI technology with human feelings and work life balance/imbalance.

The authors did a great job explaining the assumptions for the project and AI utilization. I believe the paper could be accepted for publication, after following revisions:

Thank you so much for the positive opinion and your comments! We have implemented all the changes you have suggested. 

1- Line 121 the author indicated : “We propose three components of work-family balance: time balance, involvement balance, and satisfaction balance”. I could not find the evaluation or definition of “satisfaction balance”. I recommend authors explain this factor and how they measured.

Thank you for drawing our attention to this. „We” actually referred to the authors of the quoted article, not to us, the authors of the paper. It could have been confusing. We have clarified and modified the text, so it is unambiguous. 

2- Line 238: authors indicated : “10 random workers were requested to fill in the study questionnaire:. How these 10 were selected? Voluntarily or selection by the investigators. What procedure were used in selection methods? 

We have provided some additional information on the procedure (page 13, section Dataset description). 

3- Lines 249-252, please rewrite and describe better. It is not comprehensive

Thank you for pointing that out. The sentence has been changed.

4- Lines 253-258, please simplify and explain better for a health care or HR worker, not familiar with ANN and pell ADAM.

We have added a word of explanation with regards to the ADAM optimizer and hyperparameters in general (Section: Architecture of the Machine Learning Tool, page 19).

5- Line 368: what does the “the ratio used was 9/1” mens? please explain

We have included further explanation of the subject matter and we hope this is now clear.

6- Line 392: Please provide references for algorithm selected and explain the approach 

We have added a word of explanation of the algorithm, and the reference

Once again, many thanks for the time of the reviewers, their careful reading and valuable comments. We hope that the paper can now be accepted and published, and we hope it will

---

## [Decision Letter · Decision Letter 1]

22 Apr 2020

Innovative machine learning approach and evaluation campaign for predicting the subjective feeling of work-life balance among employees

PONE-D-20-02847R1

Dear Dr. Pawlicka,

We are pleased to inform you that your manuscript has been judged scientifically suitable for publication and will be formally accepted for publication once it complies with all outstanding technical requirements.

With kind regards,

Sergio A. Useche, Ph.D.

Academic Editor

PLOS ONE

Additional Editor Comments (optional):

Reviewers' comments:

Reviewer's Responses to Questions

**Comments to the Author**

1. If the authors have adequately addressed your comments raised in a previous round of review and you feel that this manuscript is now acceptable for publication, you may indicate that here to bypass the “Comments to the Author” section, enter your conflict of interest statement in the “Confidential to Editor” section, and submit your "Accept" recommendation.

Reviewer #1: All comments have been addressed

Reviewer #2: All comments have been addressed

2. Is the manuscript technically sound, and do the data support the conclusions?

Reviewer #1: (No Response)

Reviewer #2: Yes

3. Has the statistical analysis been performed appropriately and rigorously? 

Reviewer #1: Yes

Reviewer #2: Yes

4. Have the authors made all data underlying the findings in their manuscript fully available?

Reviewer #1: Yes

Reviewer #2: Yes

5. Is the manuscript presented in an intelligible fashion and written in standard English?

Reviewer #1: Yes

Reviewer #2: Yes

6. Review Comments to the Author

Reviewer #1: Dear authors,

I'm glad to observe all the previous comments properly addressed.

At this point, I consider your text is ready for publication.

Congratulations for the work done!

Reviewer #2: This is an interesting article and innovative. The authors addressed the comments. I recommend accepting the paper for publication following the journal recommended format.

7. PLOS authors have the option to publish the peer review history of their article (what does this mean?). If published, this will include your full peer review and any attached files.

Reviewer #1: Yes: Ariel Ortiz Beltrán

Reviewer #2: No

---

## [Editor Report · Acceptance letter]

30 Apr 2020

PONE-D-20-02847R1 

Innovative machine learning approach and evaluation campaign for predicting the subjective feeling of work-life balance among employees 

Dear Dr. Pawlicka:

I am pleased to inform you that your manuscript has been deemed suitable for publication in PLOS ONE. Congratulations! Your manuscript is now with our production department. 

With kind regards,

on behalf of

Dr. Sergio A. Useche 

Academic Editor

PLOS ONE